# Optimization of PNP Degradation by UV-Activated Granular Activated Carbon Supported Nano-Zero-Valent-Iron-Cobalt Activated Persulfate by Response Surface Method

**DOI:** 10.3390/ijerph19138169

**Published:** 2022-07-04

**Authors:** Jiankun Zhang, Huifang Zhang, Lei Chen, Xiulei Fan, Yangyang Yang

**Affiliations:** 1School of Environmental Engineering, Xuzhou University of Technology, Xuzhou 221018, China; kdzh82@163.com (H.Z.); xlfan@xzit.edu.cn (X.F.); yangyy7075@126.com (Y.Y.); 2College of Energy and Environmental Engineering, Hebei University of Engineering, Handan 056038, China; xzchenlei@126.com

**Keywords:** persulfate, ultraviolet, activated carbon supported nano-zero-valent-iron-cobalt nanoparticles, response surface methodology

## Abstract

Nitrophenols are toxic substances that present humans and animals with the risk of deformities, mutations, or cancer when ingested or inhaled. Traditional water treatment technologies have high costs and low p-nitrophenol (PNP) removal efficiency. Therefore, an ultraviolet (UV)-activated granular activated carbon supported nano-zero-valent-iron-cobalt (Co-nZVI/GAC) activated persulfate (PS) system was constructed to efficiently degrade PNP with Co-nZVI/GAC dosage, PS concentration, UV power, and pH as dependent variables and PNP removal rate as response values. A mathematical model between the factors and response values was developed using a central composite design (CCD) model. The model-fitting results showed that the PNP degradation rate was 96.7%, close to the predicted value of 98.05 when validation tests were performed under Co-nZVI/GAC injection conditions of 0.827 g/L, PS concentration of 3.811 mmol/L, UV power of 39.496 W, and pH of 2.838. This study demonstrates the feasibility of the response surface methodology for optimizing the UV-activated Co-nZVI/GAC-activated PS degradation of PNP.

## 1. Introduction

Nitrophenol (NP) is a fundamental class of industrial products commonly used as intermediates, such as gunpowder, preservatives, pharmaceuticals, pigments, dyes, wood, and rubber in chemical industries [1,2,3,4]. These toxic substances pose a serious risk to human health and the natural environment and are difficult to dissipate and remove from water [5,6,7]. When some of these substances are ingested or inhaled, humans and animals are at risk of deformities, mutations, or cancer [8,9]. The U.S. Environmental Protection Agency (EPA) requires the concentrations of 2-NP, 4-NP, and 2,4-dinitrophenol (2,4-NP) in natural waters to be ≤10 μg/L as they have been classified as “priority pollutants” [10].

Traditional water treatment technologies, such as physical adsorption, membrane separation, and biological methods, generally have low removal efficiencies and high costs, making effective water treatment difficult [11,12]. Sulfate radical (SO_4_^−^) based persulfate (PS) oxidation technology (a hot research topic in recent years) has been widely used in many fields, including wastewater treatment [13,14]. SO_4_^−^, which has higher redox potential (E0 = 2.5~3.0 V) and a longer half-life than the hydroxyl radical (OH, the dominant active substance in conventional Fenton oxidation), can degrade and mineralize most organic pollutants [15]. Existing activation methods include thermal, ultraviolet (UV), alkali, and transition metal (genus (Fe, Co, Cu, Ni, Zn) activation [16,17]. Transition metal activation can effectively avoid the disadvantages of other activation methods, such as harsh reaction conditions and high energy consumption, and it is currently the simplest and most effective activation method. Transition metal activation generates SO_4_^−^ through electron transfer between low-valence metal ions and PS. Iron (Fe)-based catalysts have the advantages of low toxicity, a high geological storage capacity, and easy recovery. Commonly used Fe-based materials include nanoscale-zero-valent iron (nZVI), and ferrous oxides, Fe_3_O_4_ and Fe_2_O_3_; however, the catalytic performance of Fe_3_O_4_ and Fe_2_O_3_ is usually low [18,19,20].

Among the advanced UV-based oxidation technologies, UV/PS technology, which has been widely used in environmental pollution control research, has the advantages of a relatively low oxidant cost, stable nature, high efficiency of radical generation, fast reaction rate under mild conditions, high efficiency of organic micropollutant removal, and less likelihood of causing secondary pollution. Advanced oxidation technologies are based on metals and require less energy [21,22]. The combination of the activation characteristics of UV and transition metals is expected to achieve a higher NP degradation efficiency through lower energy consumption [23]. Therefore, this study used the UV + cobalt (Co)-nZVI/granular-activated carbon (GAC) + PS system to degrade NPs.

In this study, response surface methodology (RSM) was used to optimize the influencing factors in the process based on single-factor experiments, and a quadratic fitting model was established [24,25]. The 3D response surface plots were used to analyze the interrelationships between different experimental factors and their effects on the experimental results. The impact of the primary optimization was verified to provide a database and scientific basis for the actual NP degradation process.

## 2. Materials and Methods

### 2.1. Experimental Materials

Reagents: sodium persulfate (Na_2_S_2_O_8_), concentrated sulfuric acid (H_2_SO_4_), sodium hydroxide (NaOH), ferrous sulfate heptahydrate (FeSO_4_·7H_2_O), sodium borohydride (NaBH_4_), cobalt chloride hexahydrate (CoCl_2_·*n*H_2_O), absolute ethanol (C_2_H_6_O), and PNP. The reagents were analytically pure, methanol was chromatographically pure, and activated carbon was GAC. Reagents were purchased from Sinopharm Chemical Reagent Co.

### 2.2. Preparation of Co-nZVI/GAC

The GAC was soaked in 5% hydrochloric acid (HCl) for 24 h, washed with deionized water until the supernatant was clear, and dried in an oven at 105 °C. FeSO_4_·7H_2_O (2 g) was dissolved in deionized water, 0.05 g of polyethylene glycol was dissolved in ethanol-water solution (1:1; *v*/*v*), and 3 g of activated carbon was added to the mixture and polymerized fully by ultrasonic treatment for 2 h. The mixture was transferred to a three-neck flask and thoroughly stirred under nitrogen protection. Next, 0.03 mol/L NaBH_4_ was added to the flask and stirring was continued for 30 min. After washing with oxygen-free deionized water and absolute ethanol three times, a 0.4% cobalt chloride solution was added to the mixture and stirred for 30 min [26]. After the reaction was complete, a magnet was placed at the bottom of the three-neck flask for magnetic liquid separation, washed with oxygen-free deionized water and absolute ethanol three times, and dried in a vacuum drying oven at 75 °C to obtain Co-nZVI/GAC.

### 2.3. Comparative Analysis of Different Systems

To investigate the effects of Co-nZVI/GAC, PS, and UV on PNP degradation, individual and combined experiments were conducted for each factor. Co-nZVI/GAC was applied at 1.5 g/L, PS at 1 mmol/L, and UV power at 45 W (controlled by adjusting the number of UV lamps turned on; each lamp was 15 W). The initial pH value of the reaction was adjusted to 6 by adding 0.1 mol/L NaOH and 0.1 mol/l dilute H_2_SO_4_ solutions. The PNP concentration was 25 mg/L for all experiments.

In a conical bottle, 500 mL of PNP solution (25 mg/L) was added, and the pH was adjusted with 0.1 mol/L dilute H_2_SO_4_ and NaOH solution and rotated at 150 r/min. Quantitative Co-nZVI/GAC and PS were successively added into the conical bottle and timed; the reaction proceeded for 24 h. The water samples were filtered through (0.45 μm) and quenched by adding 1 mL of methanol. Three parallel experiments were performed for each group, and the PNP degradation rate was measured and calculated by high-performance liquid chromatography (HPLC).

### 2.4. Response Surface Optimization Experiment

The central composite design (CCD) model was used to study and optimize the factors influencing PNP degradation [27]. The effects of four independent factors (CO-nZVI/GAC dosage, PS concentration, pH, and UV power) on the PNP degradation rate were analyzed. In the experiment, the initial PNP concentration was 15 mg/L, and each factor was set at five levels: high and low axial points (code values +2 and −2), high and low cubic points (code values +1 and −1), and center points (code values 0) in a factorial design. The center point experiment was repeated six times to examine the reproducibility of the investigation, control the rationality of the model fitting, and evaluate the pure error of the experiment. All experiments were conducted in a random order to reduce system error. Design-Expert software was used to arrange the experimental combinations, analyze the results, obtain the quadratic polynomial regression equation, and conduct a regression analysis on the experimental data.

### 2.5. Analytical Methods

The crystal phase and crystallinity were analyzed using an X-ray diffractometer (XRD, D8 Advanced, Karlsruhe, Germany). The composition, content in solution, chemical state, and molecular structure of the compounds were analyzed by X-ray photoelectron spectroscopy (XPS, Escalab 250XI, Waltham, MA, USA). The iron content at different positions was analyzed using scanning electron microscopy (SEM, Quanta250/Quanta430, FEI, Hillsboro, OR, USA).

The PNP concentrations were quantified before and after the reaction by HPLC Agilent 1260, which was also used for quantitative testing [28]. The test conditions were as follows: column temperature, 30 °C; flow rate 1.0 mL/min, injection volume, 10 μL; o-NP UV detector wavelength, 279 nm; water to methanol ratio, 0.30:0.70; UV detector wavelength, 319 nm; and the proportion of water to methanol, 0.45:0.55.

## 3. Results and Discussion

### 3.1. Characterization of Materials

Figure 1 shows the SEM image of Co-nZVI/GAC. The surface of the material was evenly filled with spherical particles, which were nZVI loaded on the GAC surface. There were smaller spherical particles, which may be cobalt particles. Figure 2 shows the characteristic peaks of Fe and Co in the main body of the material, indicating that the activated carbon was successfully loaded with Fe and Co [29,30]. Figure 3 shows that 2θ diffraction peaks appeared at angles of 30.3°, 35.7°, 43.4°, and 62.8°, which corresponded to Fe_2_O_3_ (Joint Committee on Powder Diffraction Standards [JCPDS] 40-1139), Fe_3_O_4_ (JCPDS 19-0629), CoFe_2_O_4_ (JCPDS 22-1086), and FeO (JCPDS 01-1111), respectively. It also shows that Fe and Co were loaded onto the GAC.

### 3.2. Comparative Analysis of Different Systems

The degradation effects of the different combinations of systems on PNP are shown in Figure 4. PNP removal rate by a single PS system was approximately 13%, indicating that PS is relatively stable at room temperature and that it is challenging to generate free radicals.

The UV + Co-nZVI/GAC + PS system was more effective than the UV + PS and Co-nZVI/GAC + PS systems for PNP removal, indicating that the synergistic effect between Co-nZVI/GAC and UV light effectively enhanced the efficacy of sulfate radicals generated by Na_2_S_2_O_8_, which improved the PNP removal rate.

### 3.3. Establishment of Model and Response Surface Analysis

#### 3.3.1. Establishment of Model

Four factors, pH, UV power, Co-nZVI/GAC dosage, and PS concentration, were selected as the response values for PNP removal. Response surface analysis was performed using the CCD principle. The horizontal settings of the experimental variables and the CCD experimental arrangement of the response surface test are shown in Table 1 and Table 2, respectively.

Based on the results of all the CCD experiments (Table 2), a quadratic polynomial model between the Co-nZVI/GAC dosage, PS concentration, UV power, and pH and PNP removal rate was established. The *p*-value (which indicates the significance of the model) is the result obtained from the significance test and coefficients of the model in the analysis of variance (ANOVA). A *p*-value < 0.05 indicates a significant interaction effect. As shown in Table 3, the *p*-values for Co-nZVI/GAC dosing, UV power; Co-nZVI/GAC dosing, pH; and PS concentration, pH were *p* = 0.3532 > 0.05; *p* = 0.1842 > 0.05; and *p* = 0.6957 > 0.05, respectively, indicating that the interaction effects between Co-nZVI/GAC dosage, UV power; Co-nZVI/GAC dosage, pH; and PS concentration, pH were insignificant. The fitted regression equation is obtained as follows:Y = + 72.18 − 9.53A + 12.46B + 2.95C − 6.26D − 2.10AB − 0.66AC − 0.96AD − 2.99BC − 0.27BD − 2.71CD − 2.70A^2^ − 3.97B^2^ − 1.35 C^2^ − 1.67 D^2^(1)
where A is the Co-nZVI/GAC dosage, B is the PS concentration, C is the UV power, and D is the pH.

As shown in Figure 5, the points were very close to the straight line, indicating that the residual satisfies the normal distribution. Figure 6 shows the response of the actual values to the statistical model prediction. It can be observed that the experimental results have a significant correlation with the values predicted by the statistical model, further verifying the prediction model.

As shown in Table 3, the F-value of the fitted model Fisher’s test was 75.31 with a *p*-value < 0.0001, indicating that the model was excellent in predicting the PNP degradation efficacy. The misfit value was 0.5861, indicating that the model is reliable and effective. The measured signal-to-noise ratio of the model was 32.924, and the coefficient of variation (CV) was 4.27%, indicating the high accuracy of the experiment. The R^2^ values of the fitted model and the adjusted R^2^ value were 0.9860 and 0.9729, respectively. The difference between the two values was less than 0.2, indicating the rationality of the model. The *p*-values for the four factors, Co-nZVI/GAC dosage, PS concentration, UV power, and pH, were all less than 0.0001, indicating that all four elements significantly affected the response values. The order of significance was PS concentration > Co-nZVI/GAC dosage > pH > UV power.

#### 3.3.2. Response Surface Analysis

A three-dimensional response surface and two-dimensional contour map were used to analyze the interaction of the experimental factors with PNP degradation efficiency. Each map shows the interaction between the two factors with the other factors, maintained at the central level to obtain response surface plots and contour plots of the two factors on the response values. The results are shown in Figure 7, Figure 8, Figure 9 and Figure 10.

As shown in Figure 7, an increase in PS concentration and a decrease in Co-nZVI/GAC dosage improved the PNP removal rate. Transition metals and metal oxides effectively activated PS to degrade organic pollutants, and a certain amount of Co-nZVI/GAC effectively activated PS to degrade PNP. When the UV power and pH were constant, and the Co-nZVI/GAC dosage was low, the PS in the solution was not sufficiently activated to produce SO_4_^−^ (Formula (2)). An increase in Co-nZVI/GAC dosage can rapidly activate the PS in the solution in large amounts and produce excessive SO_4_^−^ [31,32]. However, at higher PS concentration, excessive S_2_O_8_^2−^ can undergo competition and disproportionation reactions (Formulas (3) and (4)) [32], which can continuously consume SO_4_^−^ and lead to a decrease in PNP degradation rate.
Fe^2+^ + S_2_O_8_^2−^ → Fe^3+^ + SO_4_^−^ + SO_4_^2−^(2)
S_2_O_8_^2−^ + SO_4_^−^ → S_2_O_8_^−^ + 2SO_4_^2−^(3)
SO_4_^−^ + SO_4_^−^ → 2SO_4_^−^ or S_2_O_8_^2−^(4)

As shown in Figure 8, increased UV power elevated the PNP degradation rate; however, the increase was not significant. For such a concentration level of PNP, a lower level of UV power can activate PS, as a continuous increase in UV power does not significantly improve the PNP degradation rate [33]. Based on the ANOVA results, the interaction term for UV power and Co-nZVI/GAC dosage did not significantly affect PNP degradation. The PNP degradation rate was higher when the Co-nZVI/GAC dosage was lower, indicating that for the UV + Co-nZVI/GAC + PS system, a lower Co-nZVI/GAC dosage combined with the synergistic effect of UV was sufficient to activate PS to produce SO_4_^−^, whereas a higher Co-nZVI/GAC dosage would cause PS to be activated rapidly and produce more SO_4_^−^ to disproportionate with excess S_2_O_8_^2−^ (Formulas (3) and (4)), and lead to a decrease in the PNP removal rate.

The interaction between UV power and PS concentration is shown in Figure 9. Increased UV power and PS concentration elevated the PNP removal rate. Increased UV power increased the input activation energy and accelerated PS activation to SO_4_^−^ [34], whereas increased PS concentration provided a richer oxidant activated to produce more SO_4_^−^ and HO, to improve the PNP removal rate.

As shown in Figure 10, the PNP removal rate was more significant at a higher UV power and lower pH. The PNP degradation rate increased with lowered pH because the pH change caused the interconversion of HO·and SO_4_^−^, and the presence of large amounts of H^+^ in the water column kept the surface of Co-nZVI/GAC highly active in deactivating Na_2_S_2_O_8_. Under acidic conditions, SO_4_^−^ is the primary free radical, and an increase in pH causes a gradual decrease in the removal rate due to the gradual accumulation of pH, which facilitates Co-nZVI/GAC surface oxidation and forms an iron hydroxide oxide layer, which affects the activation effect. However, under alkaline conditions, HO·is the primary free radical, and the oxidation potential of SO_4_^−^ is higher than that of HO (Formula (5)), resulting in different effects of pH on the PNP degradation rate [35]. A more substantial UV power has a better activation effect on PS, which facilitates the production of more SO_4_^·−^, contributing to PNP degradation.
SO_4_^−^ + −OH→ SO_4_^2−^ + HO^●^
(5)

#### 3.3.3. Model Validation

The predicted optimal experimental conditions for PNP degradation were obtained by optimizing a combination of experimental factors using Design-Expert 12 software with a Co-nZVI/GAC dosage of 0.827 g/L, PS concentration of 3.811 mmol/L, UV power of 39.496 W, and pH of 2.838. The validation test showed that the PNP degradation rate was 96.72%, which was close to the predicted value of 98.05%. This indicated that the model could better simulate the influence of various factors on the PNP removal rate.

## 4. Conclusions

A four-factor, five-level CCD-based RSM was used to evaluate the performance of the UV + Co-nZVI/GAC + PS system in PNP degradation. The effects of the independent variables (Co-nZVI/GAC dosage, PS concentration, UV power, and initial pH) and their interactions on the response factors were evaluated and analyzed using ANOVA. The initial PNP concentration was 15 mg/L, and the optimum conditions were a Co-nZVI/GAC dosage of 0.827 g/L, PS concentration of 3.811 mmol/L, UV power of 39.496 W, and a pH of 2.838. The PNP degradation rate was 96.72%, which was close to the predicted value of 98.05%. The significance and suitability of the proposed quadratic model were validated with low probabilities (˂0.0001) and high correlation coefficients (R^2^ = 0.9860), indicating a strong correlation between the predicted and experimental data. The results confirmed that RSM is a beneficial tool for optimizing the UV + Co-nZVI/GAC + PS system. Thus, the combination of the UV + Co-nZVI/GAC + PS system is an effective technique for producing reactive oxidizing radicals and efficiently degrading PNP.

## Figures and Tables

**Figure 1 ijerph-19-08169-f001:**
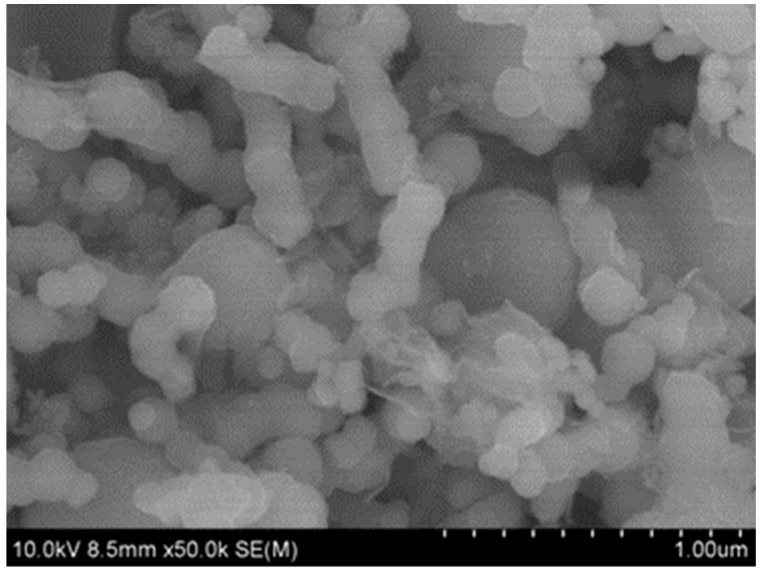
SEM of Co-nZVI/GAC.

**Figure 2 ijerph-19-08169-f002:**
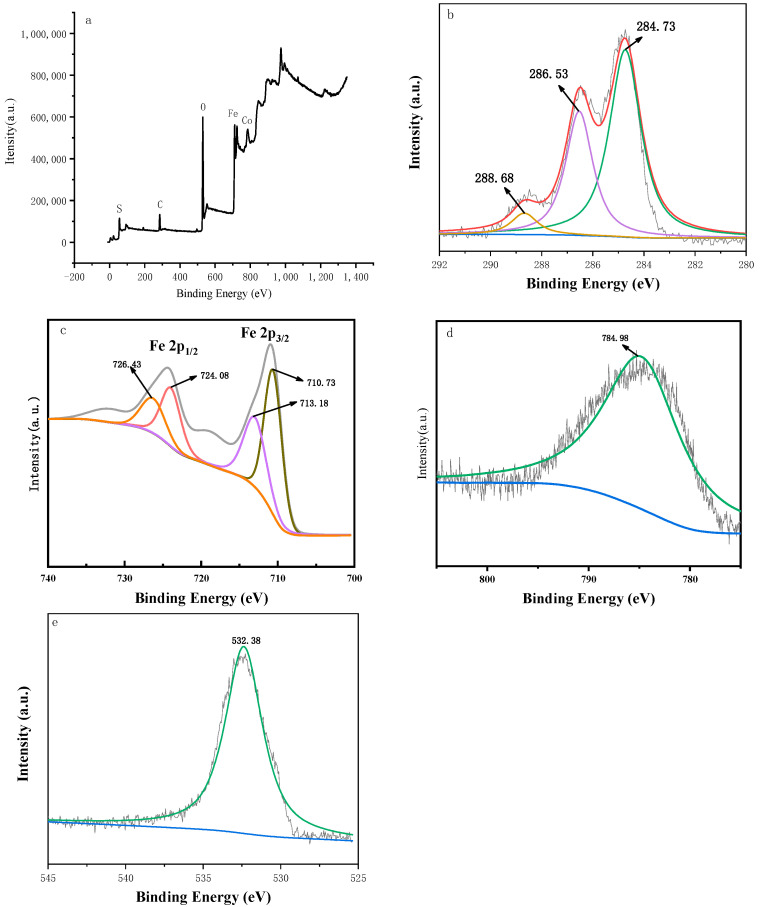
XPS spectra of Co-nZVI/GAC full scan (**a**), XPS spectrum of C in Co-nZVI/GAC (**b**), XPS spectrum of Fe in Co-nZVI/GAC (**c**), XPS spectrum of Co in Co-nZVI/GAC (**d**), XPS spectrum of O in Co-nZVI/GAC (**e**).

**Figure 3 ijerph-19-08169-f003:**
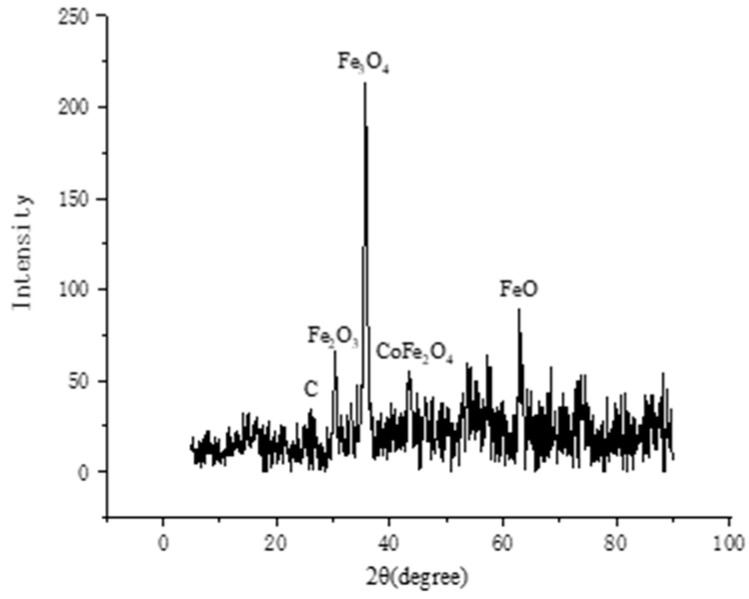
XRD spectra of Co-nZVI/GAC.

**Figure 4 ijerph-19-08169-f004:**
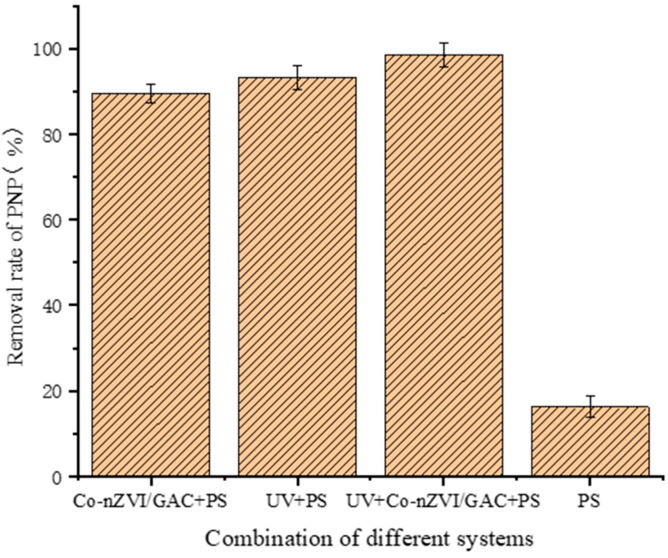
Degradation effects of different systems on PNP removal.

**Figure 5 ijerph-19-08169-f005:**
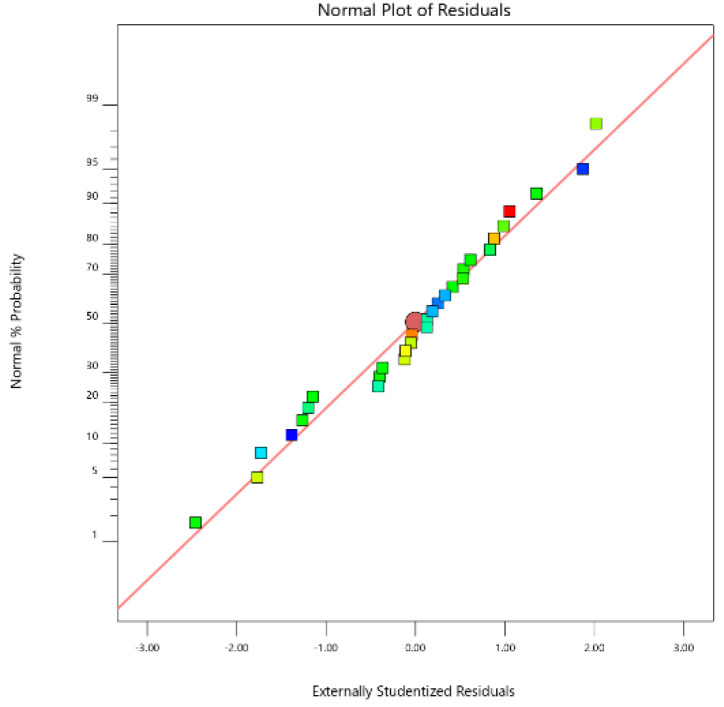
Normal probability plot of the internally studentized residuals for PNP removal.

**Figure 6 ijerph-19-08169-f006:**
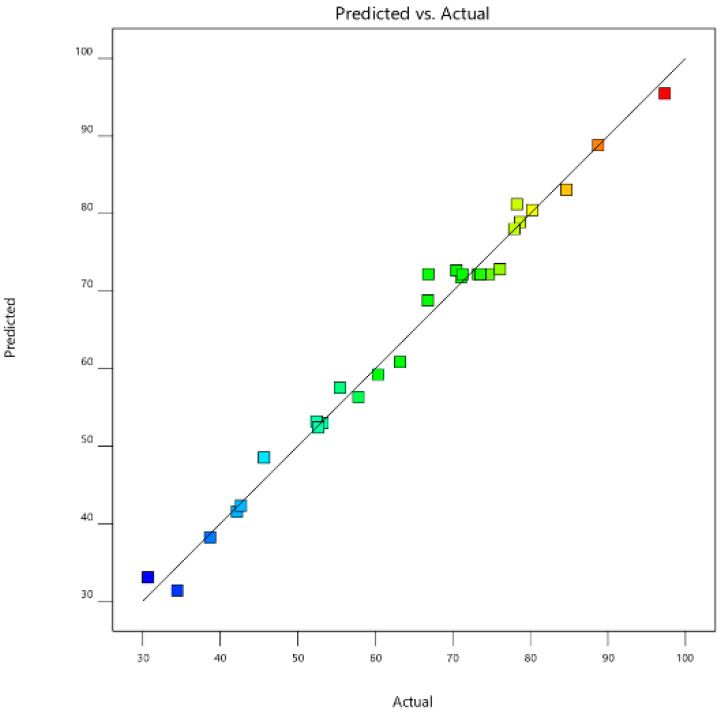
Actual values versus predicted response by CCD.

**Figure 7 ijerph-19-08169-f007:**
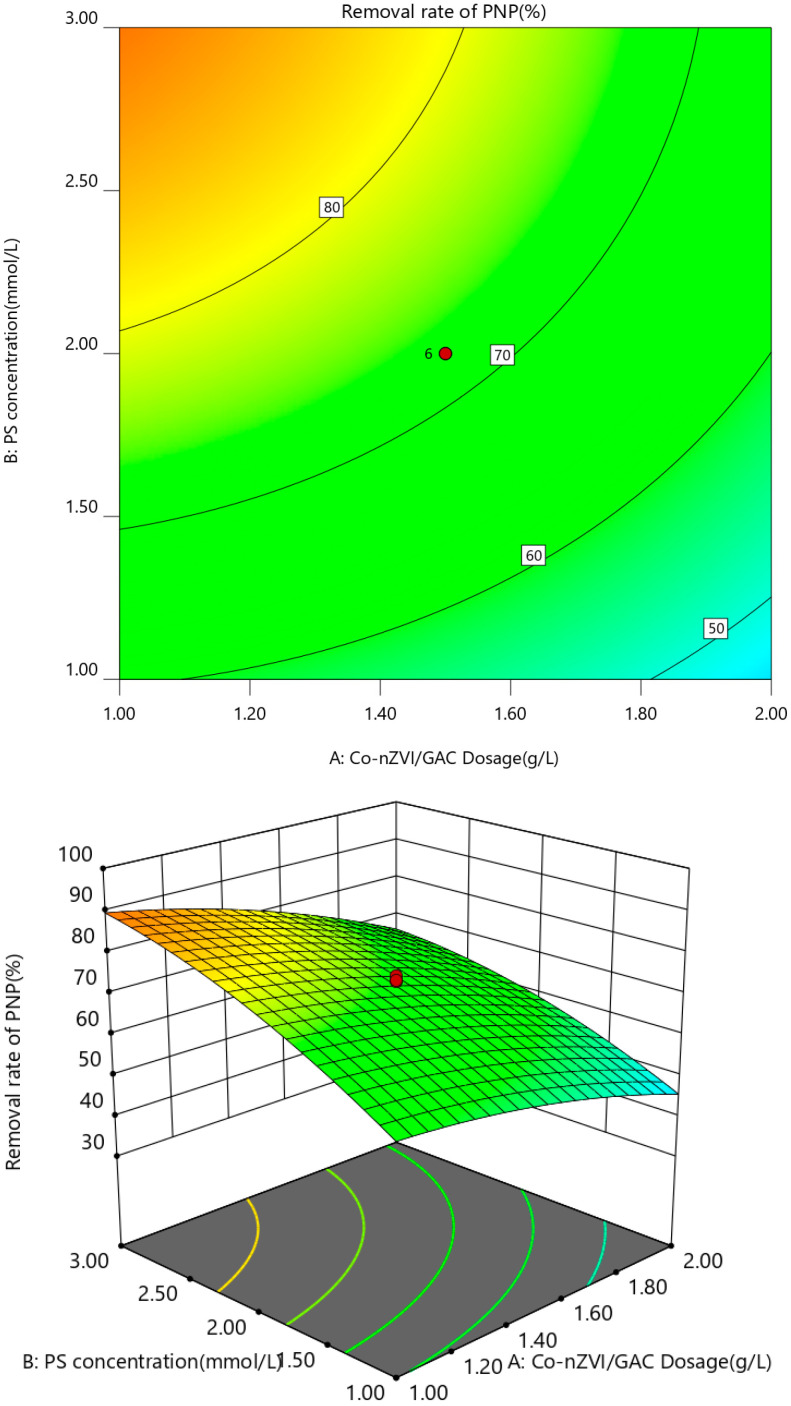
Response surface of Co-nZVI/GAC dosing and PS concentration.

**Figure 8 ijerph-19-08169-f008:**
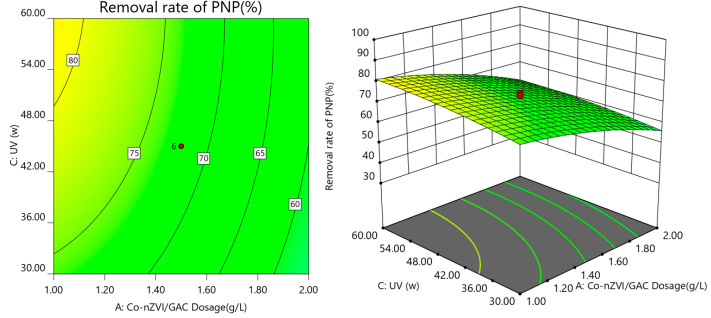
Response surface of the relationship between the Co-nZVI/GAC dosage and UV power.

**Figure 9 ijerph-19-08169-f009:**
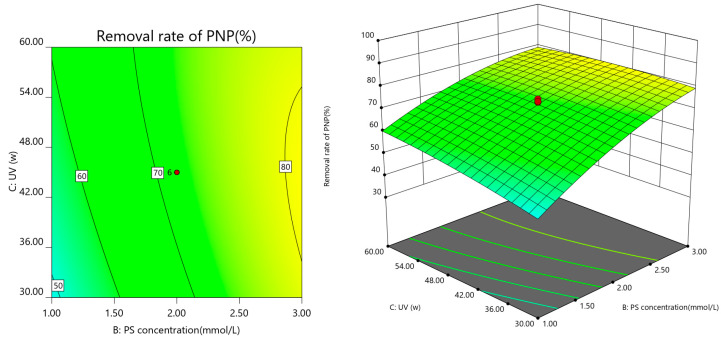
Response surface diagram of the relationship between PS concentration and UV power.

**Figure 10 ijerph-19-08169-f010:**
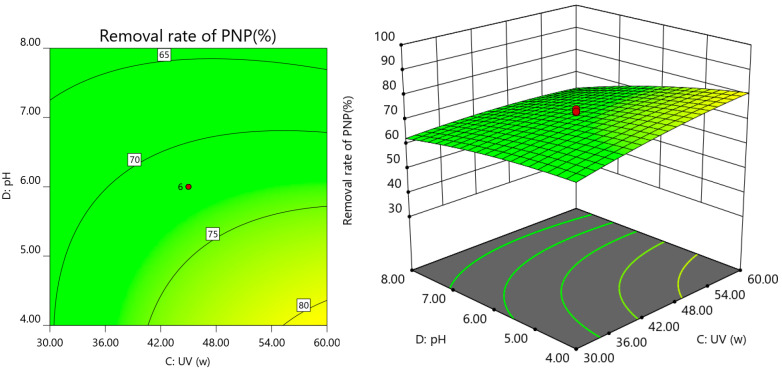
Response surface of the interaction between UV power and pH.

**Table 1 ijerph-19-08169-t001:** Design experimental factor levels.

Factor	Unit	Code	Value of Each Level				
			−2	−1	0	1	2
Co-nZVI/GAC dosage	mol/L	X_1_	0.5	1	1.5	2	2.5
PS concentration	mol/L	X_2_	0	1	2	3	4
UV power	W	X_3_	15	30	45	60	75
pH	-	X_4_	2	4	6	8	10

**Table 2 ijerph-19-08169-t002:** Test design and response values.

Test No.	Coding Variable Level				PNP Removal Rate
	X_1_	X_2_	X_3_	X_4_	
1	0	0	0	0	71.23%
2	0	0	0	0	73.56%
3	−1	1	−1	−1	88.74%
4	−2	0	0	0	80.24%
5	0	0	0	−2	77.92%
6	0	0	−2	0	63.20%
7	1	−1	−1	−1	42.16%
8	2	0	0	0	42.68%
9	−1	−1	1	−1	71.06%
10	0	0	0	0	66.83%
11	−1	−1	1	1	57.82%
12	1	1	−1	−1	66.78%
13	−1	−1	−1	−1	52.42%
14	−1	1	1	1	78.68%
15	1	1	−1	1	60.34%
16	−1	1	1	−1	97.33%
17	1	−1	−1	1	30.69%
18	0	0	0	0	73.25%
19	1	−1	1	−1	55.43%
20	1	1	1	−1	76.08%
21	0	0	0	0	73.55%
22	0	0	0	0	74.65%
23	0	0	0	2	53.21%
24	0	−2	0	0	34.46%
25	1	1	1	1	52.65%
26	−1	1	−1	1	84.65%
27	0	2	0	0	78.26%
28	1	−1	1	1	38.69%
29	0	1	2	0	70.46%
30	−1	−1	−1	1	45.65%

**Table 3 ijerph-19-08169-t003:** Regression equation coefficients and significance tests.

Source	Sum of Squares	Degrees of Freedom	Mean Square	F	*p*
Model	7989.83	14	570.70	75.31	<0.0001
X_1_	2178.37	1	2178.37	287.45	<0.0001
X_2_	3723.30	1	3723.30	491.32	<0.0001
X_3_	209.04	1	209.04	27.58	<0.0001
X_4_	940.63	1	940.63	124.12	<0.0001
X_1_X_2_	70.43	1	70.43	9.29	0.0081
X_1_X_3_	6.96	1	6.96	0.92	0.3532
X_1_X_4_	14.69	1	14.69	1.94	0.1842
X_2_X_3_	143.10	1	143.10	18.88	0.0006
X_2_X_4_	1.20	1	1.20	0.16	0.6957
X_3_X_4_	117.13	1	117.13	15.46	0.0013
X_1_^2^	199.54	1	199.54	26.33	0.0001
X_2_^2^	432.78	1	432.78	57.11	<0.0001
X_3_^2^	50.34	1	50.34	6.64	0.0210
X_4_^2^	76.58	1	76.58	10.11	0.0062
Residuals	113.67	15	7.58		
Lack of fit	73.12	10	7.31	0.90	0.5861
Pure error	40.55	5	8.11		
In total	8103.51	29			
R^2^ = 0.9860					
R^2^_adj_ = 0.9729					
Signal-to-noise ratio = 32.924					
Coefficient of variation = 4.27					

## Data Availability

The data presented in this study are available on request from the corresponding author.

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
