# Peer review of "Optimization of PNP Degradation by UV-Activated Granular Activated Carbon Supported Nano-Zero-Valent-Iron-Cobalt Activated Persulfate by Response Surface Method"

_ijerph, 2022, doi:10.3390/ijerph19138169_

Round 1

Reviewer 1 Report

The manuscript entitled "Optimization of PNP Degradation by UV-activated Granular activated carbon Supported Nano-zero-valent-iron-cobalt Activated Persulfate by Response Surface Method" tries to present a water treatment useful in the removal of p-nitrophenol. The current version of the document has serious problems that need to be addressed. The current version of the document cannot be considered for evaluation in a prestigious scientific journal. The authors propose a technology in which they use iron-cobalt particles of zero valence supported on activated carbon, however, they do not show scientific evidence in this regard.

In the Materials and Methods section, the authors do not mention the type of activated carbon used. They do not mention the analytical techniques, experimental conditions and equipment used to characterize the material. The authors do not present any scientific support that they have synthesized and used the material during their study.

The Results and Discussion section is deficient, why do the authors say they use nanoparticles? The only SEM micrograph presented (Figure 1) shows agglomerates on the order of microns. This is not a nanoparticle. The XPS spectra (Figure 2) must be presented in the S, C, O, Fe, and Co regions in high definition, and the corresponding mathematical deconvolution must be performed to obtain fundamental information. Figure 3, corresponding to the diffractogram of the material, cannot be presented like this. It lacks axes and units. The experimental conditions must be presented. Are the observed peaks experimental noise or do they correspond to crystallographic planes? Against which XRD standard is the comparison made? Where is the diffractogram corresponding to the support (activated carbon)?

All experimental development, design of experiments, and response surfaces are meaningless if the material is not shown to have been designed correctly. The manuscript should be substantially improved.

Author Response

Thank you very much for reviewing the draft. Your suggestions are very good. We seriously accept them and have revised them.

Comment  In the Materials and Methods section, the authors do not mention the type of activated carbon used. They do not mention the analytical techniques, experimental conditions and equipment used to characterize the material. The authors do not present any scientific support that they have synthesized and used the material during their study.

Respond  In ''2.1. Experimental materials'' , Added ''activated carbon is granular activated carbon''.

          In ''2.5. Analytical methods'', Added ''For analysis of wide angles (5–90◦), the crystal phase and crystallinity were analyzed using an X-ray diffractometer (XRD, D8 Advanced, Karlsruhe, Germany). The composition, content in solution, chemical state, and molecular structure of the compounds were analyzed using X-ray photoelectron spectroscopy (XPS, Escalab 250XI, Waltham, MA, USA), while the iron content at different positions was analyzed using scanning electron microscopy (SEM, Quanta250/Quanta430, FEI, Hillsboro, OR, USA). ''

Comment  The Results and Discussion section is deficient, why do the authors say they use nanoparticles? The only SEM micrograph presented (Figure 1) shows agglomerates on the order of microns. This is not a nanoparticle. The XPS spectra (Figure 2) must be presented in the S, C, O, Fe, and Co regions in high definition, and the corresponding mathematical deconvolution must be performed to obtain fundamental information. Figure 3, corresponding to the diffractogram of the material, cannot be presented like this. It lacks axes and units. The experimental conditions must be presented. Are the observed peaks experimental noise or do they correspond to crystallographic planes? Against which XRD standard is the comparison made? Where is the diffractogram corresponding to the support (activated carbon)?

Respond  1. I have changed the picture to nano scale.

    2.In Figure 2, the spectra of Fe Co c o, etc. are added

  1. In ''3.1. Characterization of materials'' , Added card information"which correspond to Fe2O3(JCPDS 40-1139), Fe3O4(JCPDS 19-0629), CoFe2O4(JCPDS 22-1086), and FeO(JCPDS 01-1111).''

Reviewer 2 Report

Dear Author(s)
The preparation of paper is good and it contains a solution of important issue (Contamination by phenolic compound), but it need moderate English checking. 

Author Response

Thank you very much for reviewing the manuscript. We have checked and revised the spelling and tense in the article.

Author Response

Thank you very much for reviewing the draft. Your suggestions are very good. We seriously accept them and have revised them.

Comment   Line 34 to 36: The author alleges that traditional water treatment listed generally have low degradation efficiency and……making it difficult to degrade them effectively….Treatment technology such as membrane separation does not degrade pollutants bur rather remove pollutants from wastewater, so listing membrane technology as one of the degrading process cannot be correct.

Respond  I have revised it“Traditional water treatment technologies such as physical adsorption, membrane separation, and biological methods generally have low removal efficiency and high cost, making it difficult to removal them effectively.” 

Comment   Line 38 to 38: Water environmental pollution treatment…this line can better be rephrased “Wastewater treatment”

Respond  I have revised it“Sulfate radical (SO4·- ) based persulfate oxidation technology, as a hot research topic in recent years, has been widely used in many fields, such as wastewater treatment.”

Comment   Line 46 to 47: The use of the hot topic again after being used in line 37 can perhaps be replaced with it has gained significant research attention or attracted researcher’s attention

Respond  Fe-based catalysts have the advantages of low toxicity, high geological storage capacity, and easy recovery.

Comment   Line 53 to 54: What does low environmental conditions mean and how is it different for mild conditions? This reviewer feels these two are referring to the same thing

Respond UV/PS technology has the advantages of the relatively low cost of oxidant, stable nature, high efficiency of radical generation, fast reaction rate under mild conditions, high efficiency of organic micropollutant removal, and less likely to cause secondary pollution,

Comment  Line 55 to 55: “…which has been widely used in various researchers”…whats the point of the part in the sentence?

Respond  which has been widely used in the research of environmental pollution control.

Comment  Line 56 to 57: “The advanced oxidation technology is based on metal (15,16). It requires less energy. Why not merge the two sentences into The advanced oxidation technology is based on metal and it requires less energy.

Respond  The content in the front part is about advanced oxidation technology based on ultraviolet (UV), and the content in the back part is about advanced oxidation technology based on metal. They are different, so they can't be written together.

Comment  Line 70 to 71: Ferrous sulfate heptahydrate is written with a capital letter F but all other chemicals with small letters, if there is no significant reason for this, this reviewer suggest that there be consistency.

Respond  I have revised it “ferrous”

Comment  Line 75 to 86: Because the authors are reporting on what was done when preparing the Co-nZVI/GAC, the tense should be past tense rather than present tense….example: The granular activated carbon was soaked in 5% HCl for 24 hours and washed with deionized water until the supernatant was clear and was dried in an oven at 105

Respond The granular activated carbon was soaked in 5% HCl for 24 hours and washed with deionized water until the supernatant was clear and was dried in an oven at 105 ℃. Weighed 2 g of FeSO4·7H2O and dissolved in deionized water, weighed 0.05 g of polyethylene glycol in ethanol-water solution (1:1; V/V), weighed 3 g of activated carbon, and added to the mixture, and polymerized fully by ultrasonic for 2 h. After that, the mixed liquid was added to the three-neck flask, thoroughly stirred under the nitrogen protection, and dropped 0.03 mol/L NaBH4 into the flask. After the dropping was completed, continue going for 30 min. Washed with oxygen-free deionized water and absolute ethanol 3 times, added 0.4% cobalt chloride solution to the solution, and continued stirring for 30 minutes[19]. After the complete reaction, put the magnet on the bottom of the three-neck flask for magnetic liquid separation, wash it with oxygen-free deionized water and absolute ethanol three times, respectively, and dry it in a 75 ℃ vacuum drying oven to obtain Co-nZVI/GAC.

Comment  Line 91 to 91: What was used to control the pH at 6?

Respond  The initial pH value of the reaction was adjusted to 6 by adding 0.1mol/l sodium hydroxide and 0.1mol/l dilute sulfuric acid solution.

Comment  Line 93 to 94: The authors are correctly presenting the paragraphs in past tense as they are reporting. But then this section is report in present tense

Respond  500 ml PNP solution (25 mg/L) was put into a brown bottle, adjusted the pH with 0.1 mol/L dilute sulfuric acid and NaOH solution, rotated at 150 r/min, and successively added quantitative Co-nZVI/GAC and PS into the conical bottle, start and time.

Comment  Line 123 and 133: In the text the image is referred to as “Fig” but in the label is labelled “Figure”. Is that correct?

Respond  This is wrong, I have revised it “Figure”

Comment  Line 124 to 125: Why not show SEM image of the GAC without the nano-zero valent iron and compare it to the one loaded?

Respond  Through the SEM photos of granular activated carbon, we believe that the surface of activated carbon without nano zero-valent iron particles is smooth, and there is no attachment on the surface, so there is no comparison.

Comment  Line 126 to 126: Fig 1 is referred to as “Fig” and then the second image as “Figure”… why not be consistent?

Respond  I have revised it “Figure”

Comment  Line 139 to 139: Also, image 3 is referred to as “Fig”

Respond  I have revised it “Figure”

Comment  Line 137, 147 and 178: All those figures are labelled “Figure 3”… is that correct?

Respond  That not correct.I have revised the figure number.

Comment  Line 185 to 186: The R2 can be corrected to R2 

Respond   I have revised it “R2

Comment  Line 281 to 335: The reference list from number 2 seems to be double numbered

Respond   I have revised the figure number.